# Predictive Models of Double-Vibropolishing in Bowl System Using Artificial Intelligence Methods

**Joselito Yam II Alcaraz [1], Kunal Ahluwalia [2] and Swee-Hock Yeo [1,\*]**

[1] Mechanical and Aerospace Engineering, Nanyang Technological University, Singapore 639798, Singapore; yam.alcaraz@gmail.com

[2] Rolls-Royce@NTU Corporate Laboratory, Singapore 637460, Singapore; kunalahluwalia@yahoo.com

\* Correspondence: mshyeo@ntu.edu.sg; Tel.: +65-67905539

**Abstract:** Vibratory finishing is a versatile and efficient surface finishing process widely used to finish components of various functionalities. Research efforts were focused in fundamental understanding of the process through analytical solutions and simulations. On the other hand, predictive modelling of surface roughness using computational intelligence (CI) methods are emerging in recent years, though CI methods have not been extensively applied yet to a new vibratory finishing method called double-vibropolishing. In this study, multi-variable regression, artificial neural networks, and genetic programming models were designed and trained with experimental data obtained from subjecting rectangular Ti-6Al-4V test coupons to double vibropolishing in a bowl system configuration. Model selection was done by comparing the mean-absolute percentage error and r-squared values from both training and testing datasets. Exponential regression was determined as the best model for the bowl double-vibropolishing system studied with a Test MAPE score of 6.1% and a R-squared score of 0.99. A family of curves was generated using the exponential regression model as a potential tool in predicting surface roughness with time.

**Keywords:** vibratory finishing; double vibro-polishing; artificial intelligence; regression; neural network; genetic programming

---

## 1. Introduction

Vibratory finishing is a popular and established mass finishing process due to its versatility, as evidenced by the process's ability to clean, deburr, descale, and edge radius. The process is versatile enough that it can be used in polishing a large scale of materials—from removing large burrs in metallic workpieces to removing microscale roughness in plastic media.

In the aerospace industry, vibratory finishing has been used extensively because of strict requirements on surface features. Critical parts in aerospace with free-form surfaces, such as blisks, turbine blades, and fan blades benefit from this process of loose abrasive polishing.

Research in vibratory finishing inclines towards optimization through experimentation. Optimized process frameworks are commonly derived through empirical trial and error methods due to the complexities imposed by the process and its underlying principles. As a result, several attempts have been made to develop analytical models relating to various key process indicators.

Modelling these kinds of systems has a high degree of complexity due to the nature of the abrasive flow in the media. Thus, models are created to roughly predict what happens to a workpiece after putting it in a vibratory bowl, while observing process parameters such as vibration frequency and amplitude. The models created must accurately predict the output variables as well as capture the dynamics of the system. The research community coined this term as generalization modelling—a model created from

data aimed to generalize and predict future outcomes. Better generalization capability of a model indicates that it has mostly captured the physics and dynamics of a system.

The fact that the use of Artificial Neural Networks (ANN) in several machining processes has shown its potential in predictive modelling [1–3] is the main motivation of this work. ANN may also have an advantage when used in realistic environments, e.g., in an actual factory, where data would be sparse and not experimentally designed.

Furthermore, genetic algorithms and their derivatives have also been gaining traction in vibratory finishing research [4–6]. The various experimental research on the study of effects of different process parameters resulted in an abundance of data. With sufficient data and the right modelling tools, a dependable and generalized predictive model can be created.

This study aims to derive an empirical model to predict surface roughness of a workpiece subjected to a vibratory finishing process. Moreover, a relatively new class of fixture vibratory finishing process called double vibro-polishing [7], has reported to produce significant improvements with cycle times. The abundance of experimental data has potential to be used in numerical modelling to produce a versatile model that could be used by industry to save time on experiments as well as catalogue various components against the parameters for vibratory finishing. This study attempts to derive and evaluate numerical models using different methods, i.e., regression, artificial neural networks, and genetic algorithms, to establish a suitable tool for predicting cycle times with double-vibratory finishing.

Supplementing this introduction is a review of the previous trends and current state-of-the-art in analytical, numerical simulation, and computational intelligence models for the vibrofinishing process. Sections 2.1–2.7 will be a presentation of the basic building blocks of the modelling methodologies used, which would also include model conditioning and data-preprocessing. The rest of the methodology section discusses the models created and the specific parameters that were used.

A summary and comparison of the MAPE and R-squared scores of the modelling methodologies for the bowl vibratory finishing system is discussed in Section 3, where the best model is chosen. The selected model is then used to create a family of curves using a defined set of input variables. Finally, a short conclusion of the study and suggestions for future work are discussed in Section 4.

*1.1. Double Vibro-Polishing*

A vibratory bowl finishing system consist of a vertical shaft, in which the eccentric weights induce a rolling and feeding motion of media particles [8]. The schematic diagram of the vibratory bowl is shown in Figure 1.

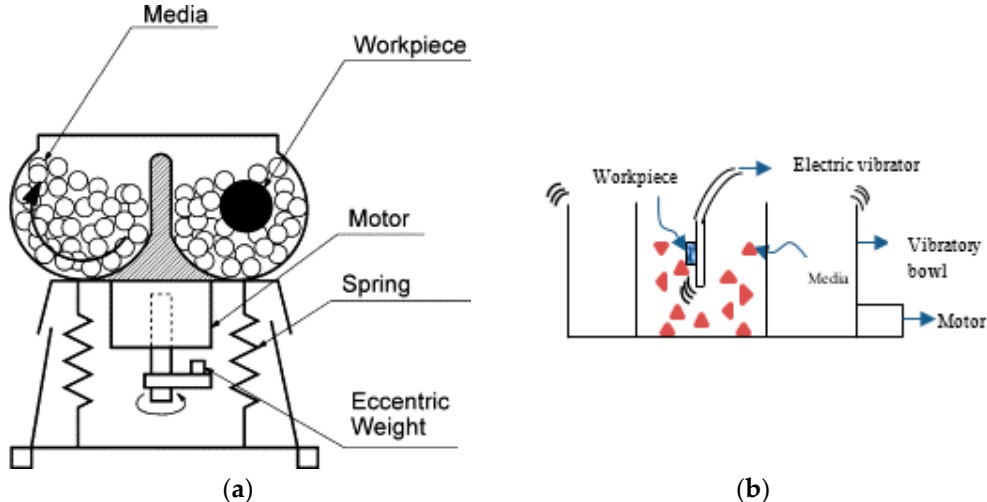

(**a**)　　　　　　　　　　　　　　　　　　　　　　　(**b**)

**Figure 1.** (**a**) Schematic Diagram of conventional Vibropolishing in bowl system [9]; (**b**) Schematic of Double Vibropolishing [7].

In most conventional vibratory machining processes, components are either freely flowing together with the vibratory media particles or held into place in a stationary fixture. Ahluwalia et al. [7] introduced a new vibropolishing method, coined double-vibropolishing. Instead of a stationary fixture, vibration was introduced to the fixture to increase the relative motion of particles to the workpiece. The fact that both the fixture and the reservoir are both vibrating suggest that higher impact forces between the media and workpiece result to more aggressive mechanical working of the workpieces.

Their experiments suggest that using the vibratory fixture in "active", i.e., the fixture is vibrating, causes a steeper drop in $R_a$ values compared with the static fixture, which then causes a 75% decrease in time to reach the cut-off roughness in the Ti-6Al-4V workpieces as shown in Table 1. As both the container and fixture are vibrating, the cycle times are improved because of higher relative motion between media and workpiece. Ti-6Al-4V was used as the experiment material because of its widespread use in aerospace industry.

**Table 1.** Ti-6Al-4V chemical composition [10].

| Carbon (Maximum) | 0.10 | % | Titanium | Balance | |
|---|---|---|---|---|---|
| Aluminum | 5.50 to 6.75 | % | Vanadium | 3.50 to 4.50 | % |
| Nitrogen | 0.05 | % | Iron (Maximum) | 0.40 | % |
| Oxygen (Maximum) | 0.020 | % | Hydrogen (Maximum) | 0.015 | % |
| Other, Total (Maximum) | 0.40 | % | | | |

## 1.2. Analytical and Mathematical Modelling

An extensive literature review reveals that several predictive models exist to study force interactions, material removal rates, and residual stresses. One of the earliest predictive modelling studies in vibratory finishing was done by Sofronas and Taraman [11], where predictions of height reduction, edge radiusing, and surface finish were empirically modelled as functions of time, media size, vibratory frequency, and workpiece hardness. It was found out that vibratory frequency is the most significant contributor to the dependent variables studied, next to media size and processing time.

Hashimoto [12] established fundamental principles and rules of vibratory finishing based on derivation of equations from experimental results. In his research, his rules include: (1) defining roughness limitation, (2) the exponential nature of roughness change, and (3) that there was a constant stock removal rate. The model developed using these principles accurately predicts surface roughness and stock removal from experiments, allowing the computation of an optimal cycle time given an initial roughness of the material.

Domblesky et al. [13] similarly developed a model that describes material removal rate as a function of bowl acceleration, workpiece mass, velocity, and specific energy. Their study reinforces the conclusion the paper of Hashimoto [12] that material removal is essentially constant over time as the finishing process approaches steady state condition.

## 1.3. Computational Intelligence Methods

Further efforts in formulating numerical solutions to mass finishing processes, or machining processes in general, is covered by artificial intelligence or computational intelligence techniques. This class of solutions works by developing a numerical relationship between input or independent parameters of the process to the observed and measured output or dependent variables.

Feng and Wang [14] developed empirical models of surface roughness, as workpieces are subjected to finish turning. Their study compared a regression model and an Artificial Neural Network (ANN) model of a turning process, with the surface roughness dependent on machining parameters, i.e., workpiece hardness, feed rate, tool radius, and cutting speed. Statistical tests were carried out to compare the goodness of fit for both regression and ANN models. Their results suggest that both models satisfy goodness of fit criteria during model development and training. Moreover, both of

these models show satisfactory prediction to additional experimental data. Their method provides a valuable tool in modelling similar applications in manufacturing engineering and design.

Other non-conventional modelling techniques were also explored in literature. A hybrid Genetic Programming - Artificial Neural Network (GP-ANN) model was designed by Garg and Tai [4] to model a vibratory finishing process. Genetic programming works by automatically evolving primitive mathematical operations using reproduction and/or crossover operations iteratively until a fitness function is satisfied [15]. Genetic Expression Programming (GEP), a variant of Genetic Programming, was used by Vijayaraghavan and Castagne [16] to create a new numerical model from the vibratory finishing data of Domblesky et al. [13]. GEP was used in combination with ANN to formulate a model to predict the power consumption and mass removal rate of the finishing process. The results of their study turned out to be comparable to the original model by Domblesky et al. [13].

In another study, Vijayaraghavan and Castagne [5] used a hybrid Multi-Adaptive Regression Splines and Genetic Programming (MARS-GP) method to predict finish condition under uncertain input parameters. The data that they used was from the study of Sofronas and Taraman [11]. Sensitivity and parametric analyses were done to identify the dominant input variables, which are processing time, media size, and vibration frequency, in that order.

## 2. Materials and Methods

### 2.1. Regression

The solution used for the regression model was to use non-linear least squares, which uses an iterative solution to find the coefficients of a non-linear function. The nonlinear function is estimated by minimizing the sum of squared errors, i.e., $\sum_{i=0}^{n} (E^2) = \sum_{i=0}^{n} (Y_i - \hat{Y}_i)^2$, using the Levenberg–Marquart algorithm, or Damped Least-Squares as used by the curvefit tool from the Python module, scipy.

### 2.2. Artificial Neural Network

In all of the experiments that are studied in this thesis, the sole dependent variable is the surface roughness $R_a$, thus, the ANN architecture used was a fully-connected, single hidden layer with n nodes, and a scalar valued output $R_a$ as shown in Figure 2. The activation function from the input layer to the hidden layer is the sigmoid function as in Equation (1) [17], while the activation function from the hidden layer to the output layer is a linear function Equation (2) to produce a continuous output variable $R_a$.

$$S(x) = \frac{1}{1 + e^{-x}} \tag{1}$$

$$L(x) = x \tag{2}$$

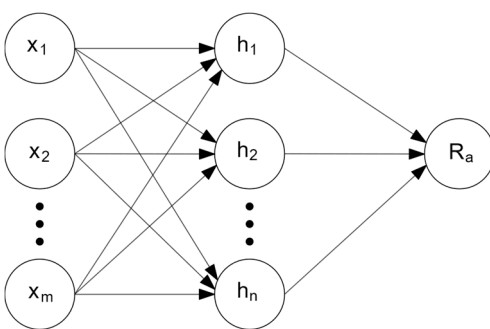

**Figure 2.** General fully connected single layer Artificial Neural Network ANN architecture.

### 2.3. Genetic Programming

Genetic programming works in a way where computer programs or structures are evolved to solve a specific problem, by evaluating the program's "fitness", or a metric to gauge its ability to

solve the problem. The basic concept is that it creates more complicated equations from a set of simpler functions. For example, the "seed" functions may include addition, subtraction, multiplication, negation, etc. Initially, an initial population of expressions are generated from these seed functions and the input variables. These initial population of expressions are evaluated using an error function, e.g., root mean squared error (RMSE), where the best models are determined. The algorithm will do random genetic operations, such as crossover and subtree-mutation, and produce a next generation of expressions. This process repeats for a specified number of generations, or with a specified termination criterion, e.g., RMSE is lower than a specified error threshold value.

### 2.4. Model Conditioning

Achieving good performance during model training requires preconditioning of input and output data. A typical preconditioning method is normalization, which is the scaling of target data to values in the range of $0 \leq \hat{y} \leq 1$, or $-1 \leq \hat{y} \leq 1$. Previous studies in vibratory finishing normalize the surface roughness by dividing the roughness value of a point in time by the initial roughness [6], and thus

$$\hat{R}_a = \frac{R_a(t)}{R_i} \tag{3}$$

This makes the model scaled in the range $1 \geq \hat{R}_a > 0$. However, it is noted that at $t \rightarrow \infty$, the roughness average will saturate to a roughness value, $R_s$, depending on the workpiece and finishing media materials. The normalized values are then actually only in the range $1 \geq \hat{R}_a > \frac{R_s}{R_i}$.

Another way to normalize the roughness average is by utilizing both the initial roughness, $R_i$, and roughness saturation, $R_s$ in such way as follows:

$$\hat{R}_a = \frac{R_a(t) - R_s}{R_i - R_s} \tag{4}$$

This formulation allows the normalized roughness values to be within the range $1 \geq \hat{R}_a > 0$, as well as approach zero as $t \rightarrow \infty$. This is superior to Equation (3) since the first method will not approach roughness values of zero with time. Different saturation values—even in normalized space—would tend to require an unnecessarily complex model to achieve a good match, and is almost the same as not doing normalization at all. Scaling the data using Equation (4) reduces the degrees of freedom needed for the model, in such a way that the starting normalized $R_a$ value is always 1, and will always approach zero with time, as with the definition of initial roughness and roughness saturation.

### 2.5. Acquisition and Pre-Processing of Data

Similar to the experiments done by Ahluwalia et al. [7], Ti-6Al-4V workpieces were subjected to double-vibropolishing in a vibratory bowl to study several factors that affect the average surface roughness. Specifically, the independent variables studied were:

- time subjected to finishing, $t$,
- initial surface roughness, $R_i$,
- frequency of vibratory bowl, $f_b$,
- state of vibratory fixture, $s_v$

with the average roughness as the only independent variable $R_a$.

### 2.6. Feature Generation

There are four independent variables that were investigated. Thus, there are a total of $2^4 - 1$ variables including the base variables, ($t$, $f_b$, $R_i$, $s_v$), and the interaction terms. Table 2 shows all of the generated terms. These terms are used as the features in creating the regression and ANN models.

**Table 2.** Base and interaction variables for vibratory bowl modelling.

| Time ($t$) | Bowl Frequency ($f_b$) | Initial Roughness ($R_i$) | Vibratory Fixture State ($s_v$) | Combined Terms |
|---|---|---|---|---|
| 0 | 0 | 0 | 1 | $(s_v)$ |
| 0 | 0 | 1 | 0 | $(R_i)$ |
| 0 | 0 | 1 | 1 | $(R_i)(s_v)$ |
| 0 | 1 | 0 | 0 | $(f_b)$ |
| 0 | 1 | 0 | 1 | $(f_b)(s_v)$ |
| 0 | 1 | 1 | 0 | $(f_b)(R_i)$ |
| 0 | 1 | 1 | 1 | $(f_b)(R_i)(s_v)$ |
| 1 | 0 | 0 | 0 | $(t)$ |
| 1 | 0 | 0 | 1 | $(t)(s_v)$ |
| 1 | 0 | 1 | 0 | $(t)(R_i)$ |
| 1 | 0 | 1 | 1 | $(t)(R_i)(s_v)$ |
| 1 | 1 | 0 | 0 | $(t)(f_b)$ |
| 1 | 1 | 0 | 1 | $(t)(f_b)(s_v)$ |
| 1 | 1 | 1 | 0 | $(t)(f_b)(R_i)$ |
| 1 | 1 | 1 | 1 | $(t)(f_b)(R_i)(s_v)$ |

*2.7. Data Extrapolation Using Exponential Function*

Since the roughness saturation was established for the bowl experiments, an exponential function can now be fit to individual coupons. The creation of additional points for the model to learn from makes the training process stable for all models.

The datapoints were fitted with an exponential curve of the form:

$$R_a = e^{a_0} e^{a_1 t} + R_s \tag{5}$$

where $R_a$ is the roughness average, $R_s = 0.12$ µm is the estimated roughness saturation, and $a_0$ and $a_1$ are the approximated values from curve fitting.

Figure 3 shows the individual exponential fits for all of the bowl experiments. It can be seen that the roughness value for all of the coupons approaches the saturation value of $R_s = 0.12$ µm as it is subjected to longer finishing time. These exponential fits are then used as the inputs to all the subsequent models, i.e., regression, ANN, and GP.

*2.8. Prediction Models*

2.8.1. Regression Model

An exponential regression model was created using the terms from Table 2 and was reduced to eight terms by selection of significant terms. The terms in Table 3 show that the only non-interaction term was the time, $t$, and the other terms $f_b$, $R_i$, and $S_v$ affect the model through interaction with $t$, including 3rd and 4th order interactions between the variables.

Figure 4 shows the predicted $R_a$ vs. measured $R_a$ plot in the normalized, $\hat{R}_a$ space, where all values are between zero and one. The diagonal green line is the ideal line where the predicted value is exactly the same as the measured value. This gives a visual indication of how well the model fits the data. Note here that the values used for training are the normalized $\hat{R}_a$ values, which are obtained using the method discussed in Section 2.4, Model Conditioning.

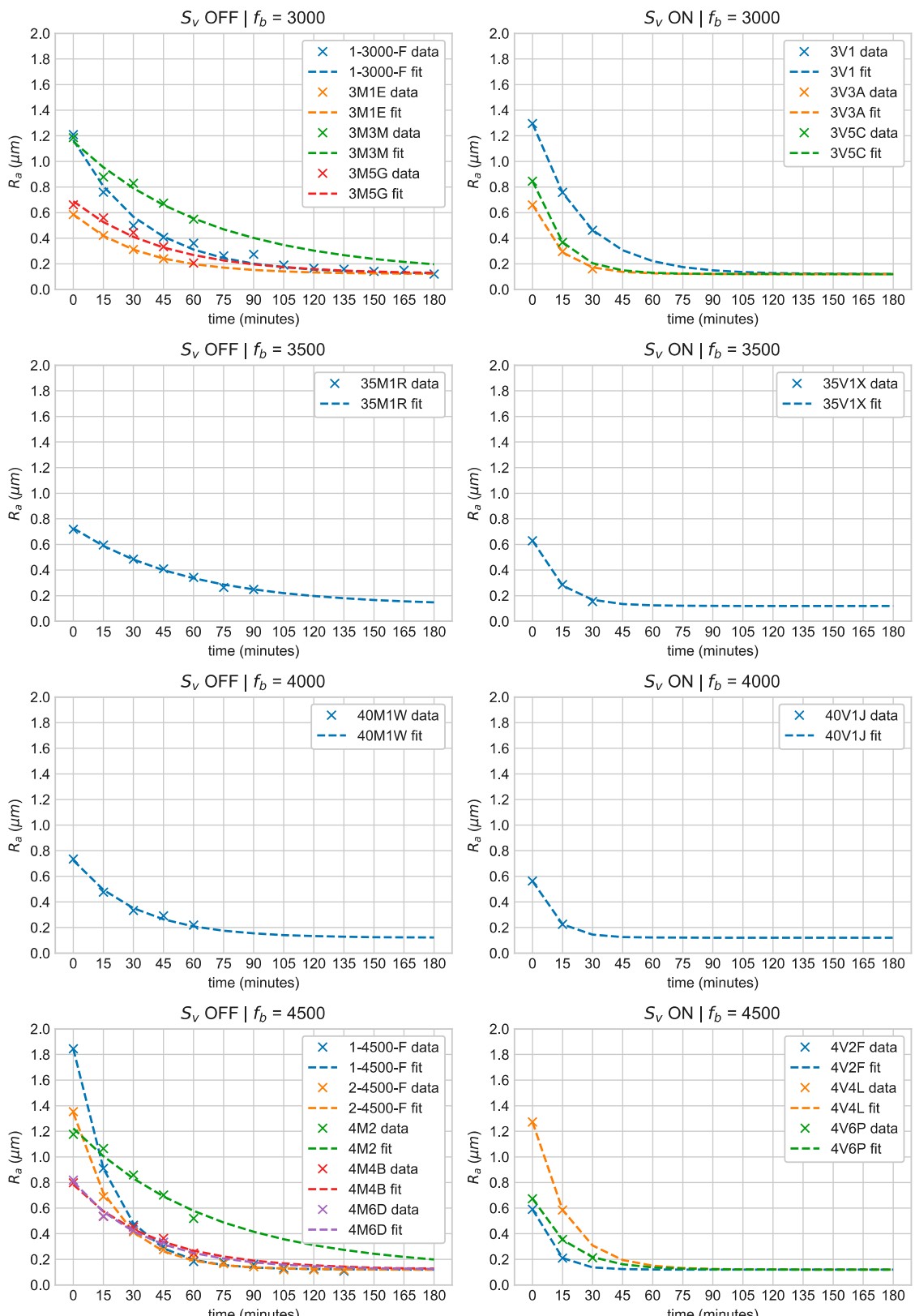

**Figure 3.** Data for vibratory bowl, $S_v$ is the secondary vibratory fixture state, $f_b$ is the bowl frequency.

**Table 3.** Significant variables for exponential regression.

| Terms | Coefficient | Standard Error | t | P > |t| |
|:---:|:---:|:---:|:---:|:---:|
| $(t)$ | $-13.15$ | 0.13 | $-103.41$ | 0 |
| $(t)(f)$ | 11.65 | 0.14 | 81.12 | 0 |
| $(t)(R_i)$ | 24.55 | 0.24 | 103.03 | 0 |
| $(t)(S_v)$ | $-45.29$ | 0.27 | $-167.16$ | 0 |
| $(t)(f_b)(R_i)$ | $-32.42$ | 0.26 | $-125.05$ | 0 |
| $(t)(f_b)(S_v)$ | 34.33 | 0.3 | 113.14 | 0 |
| $(t)(R_i)(S_v)$ | 40.71 | 0.33 | 122.31 | 0 |
| $(t)(f_b)(R_i)(S_v)$ | $-32.95$ | 0.37 | $-90.14$ | 0 |

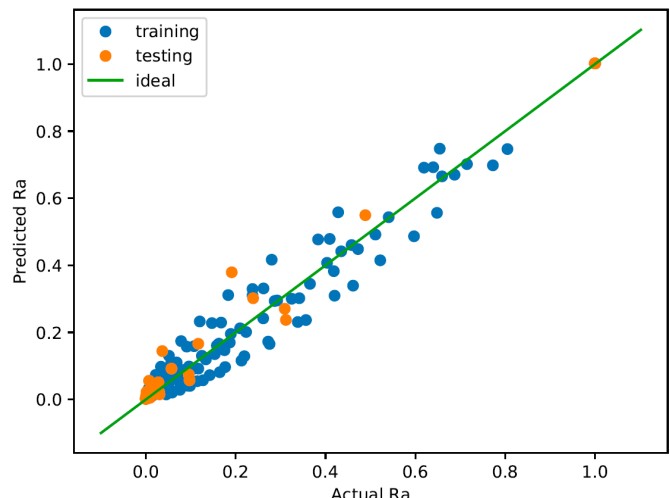

**Figure 4.** Exponential regression predicted vs. measured (normalized values) for bowl system.

### 2.8.2. Artificial Neural Network

The inputs for the ANN model are the same terms as in the ones generated in Table 2. Grid-search was done to optimize the hyper-parameters of the ANN model and are summarized in Table 4.

**Table 4.** Optimized Artificial Neural Network (ANN) Parameters for bowl system.

| Parameter | Value |
|:---:|:---:|
| Number of hidden layers | 1 |
| Number of nodes in hidden layer | 7 |
| Learning rate | 0.03 |
| Regularization constant | 0.0 |
| Total Epochs | 5320 |

The results of training are shown in the predicted vs. measured $\hat{R}_a$ plot in Figure 5. The training datapoints generally follow the ideal curve well, with the exception of the points at $\hat{R}_a = 1$, which is the initial $R_i$.

### 2.8.3. Genetic Programming

Finally, a GP model was also created for the bowl system experiments. Similar with the trough system, the inputs for the GP model were the base independent variables only, which are time, $t$, initial roughness, $R_i$, bowl frequency, $b_f$, and vibratory fixture state, $S_v$. Grid-search was again used to optimize the hyperparameters of the GP model. The optimized parameters are shown in Table 5.

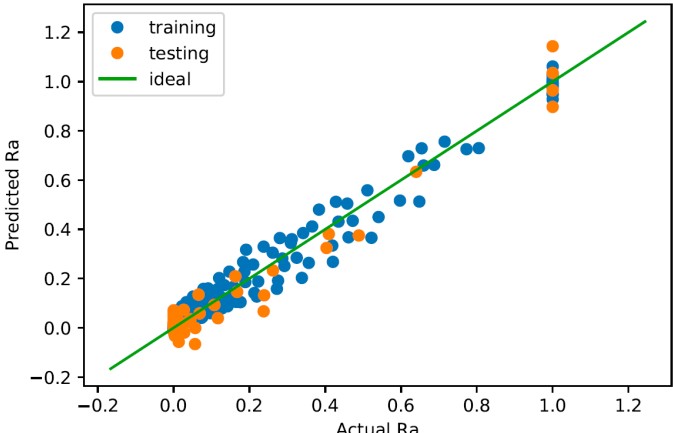

**Figure 5.** Artificial Neural Network (ANN) predicted vs. measured (normalized values) for bowl system.

**Table 5.** Genetic programming (GP) model optimized hyperparameters for bowl system.

| Parameter | Value |
|---|---|
| Parsimony coefficient | $4 \times 10^{-5}$ |
| Generations | 20 |
| Crossover probability | 0.9 |
| Subtree-mutation probability | 0.05 |
| Population size | 5000 |
| Stopping criterion | $|\bar{e}| < 0.01$ |
| Function set | (add, subtract, multiply, divide, negate, exponential) |

Similar with the regression and ANN models, the training and testing results of the GP model in the normalized space is shown in Figure 6. The GP model was able to fit the training data and also generalized to test data very well. A moderately complex expression tree was generated for the bowl system as shown in Figures 7 and 8.

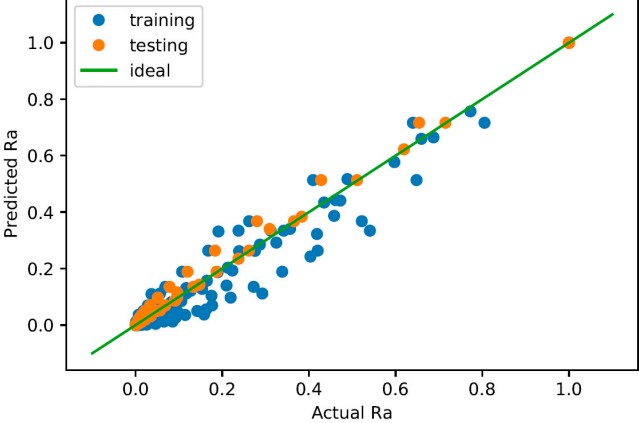

**Figure 6.** Genetic Programming predicted vs. measured (normalized values) for bowl system.

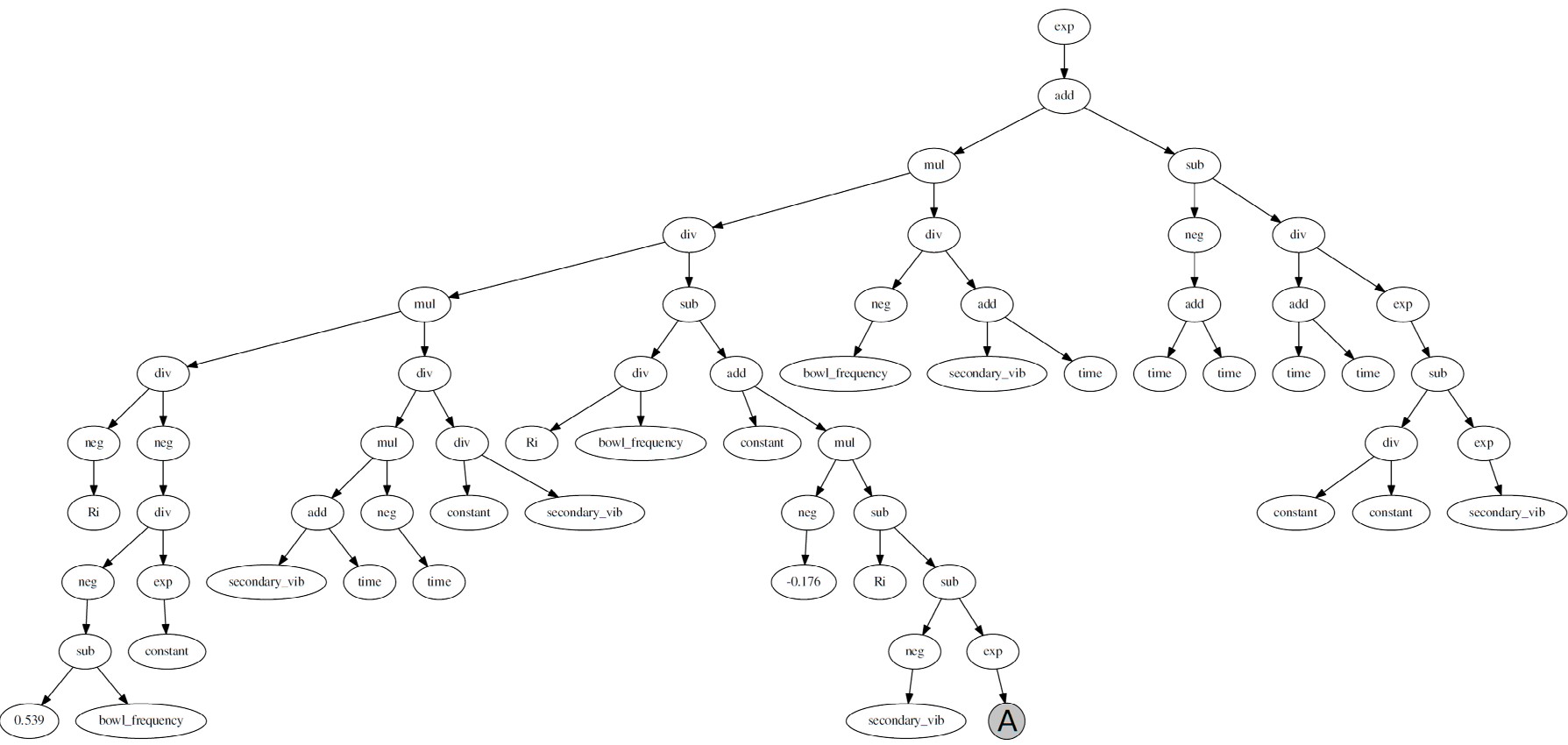

**Figure 7.** Program tree generated through genetic programming for bowl system.

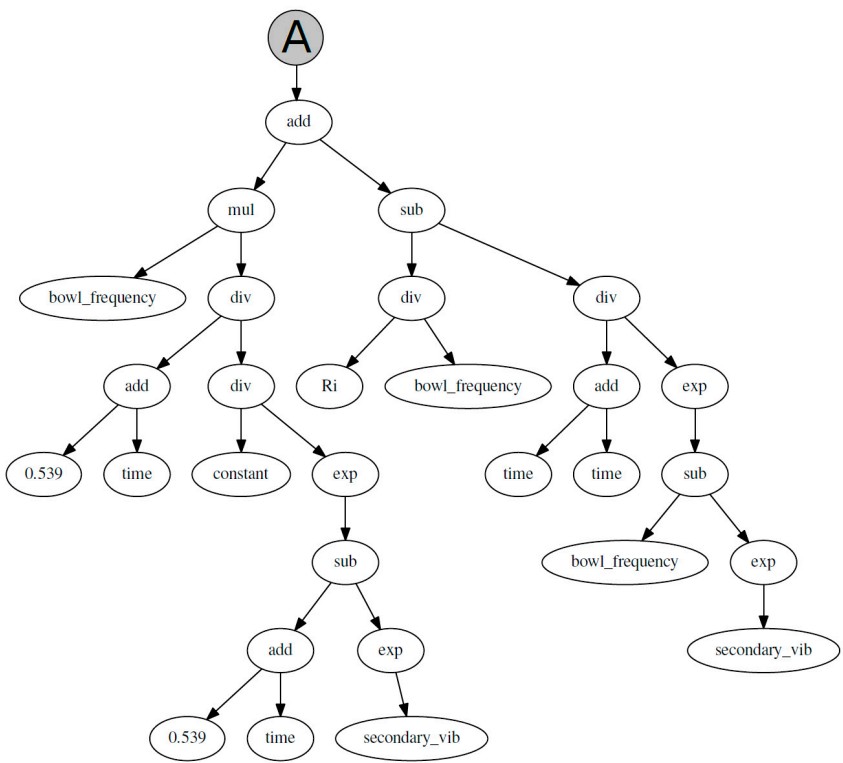

**Figure 8.** Program tree generated through genetic programming for bowl system, continued from Figure 7.

## 3. Discussion of Results

### 3.1. Model Selection

Putting all the modelling efforts into perspective, the summary of all the results for the different models and vibratory finishing experiments are summarized in Table 6.

**Table 6.** Summary of results for bowl system.

| Model Type | Train MAPE (%) | Train R-Squared | Test MAPE (%) | Test R-Squared |
|---|---|---|---|---|
| Exponential regression | 8.0 | 0.98 | 6.1 | 0.99 |
| ANN | 9.8 | 0.97 | 15.4 | 0.96 |
| GP | 7.4 | 0.97 | 5.3 | 0.98 |

The bowl system model scores were computed using the extrapolated points determined and as described in Section 2.7, since the number of raw data points were not consistent for each coupon, i.e., the cut-off times were different for each of the coupons, depending on the roughness values. These extrapolated points are used as the "measured or true values" which is compared with the model generated values to compute the training and testing scores. The scores are then used as basis to select the best model from the three models discussed in this study.

Though the R-squared test scores show good results for all the models, it can be concluded that for all the methods explored for the bowl experiments, exponential regression performed best. The GP model would have a lower score in terms of MAPE, but preference will still be on the exponential regression model because of the lower level of complexity and better interpretability. Thus, the exponential regression model will be used to create a family of curves which serves as a potential tool in predicting surface roughness with time.

*3.2. Family of Curves Generation*

In the interest of creating a double-vibropolishing prediction tool for bowl systems, a family of curves was generated using the exponential regression model. Of course, the input variables that were studied are the input variables used in generating the curves, and the domain for each parameter is maintained to limits of the experimental data. As such, the parameters used are as follows:

- time, $t$, are 15 min intervals from 0 to 180 min
- bowl frequencies, $b_f$, are 3000, and 4500 rpm
- secondary vibratory fixture state of 0 for "off" and 1 for "on", and
- initial roughness, $R_i$ of 5 equally spaced points from 0.6 to 1.5 μm

Following the model trends in the generated curves shown in Figure 9, it can be inferred that the surface roughness reduction time decreases if the bowl frequency is increased from 3000 rpm to 4500 rpm, and also if the secondary vibratory fixture is turned on. This is consistent with the observation of Ahluwalia et al. [7] where the vibrating fixture helps in finishing the workpieces in a faster rate, thus it can be used a valid model.

While the family of curves are generated using a finite set of parameters, it should be noted that these are variable and can be set arbitrarily as long as the values are still within the simulation domain. The intervals set are determined by the authors as the ideal for visualization and interpretation purposes.

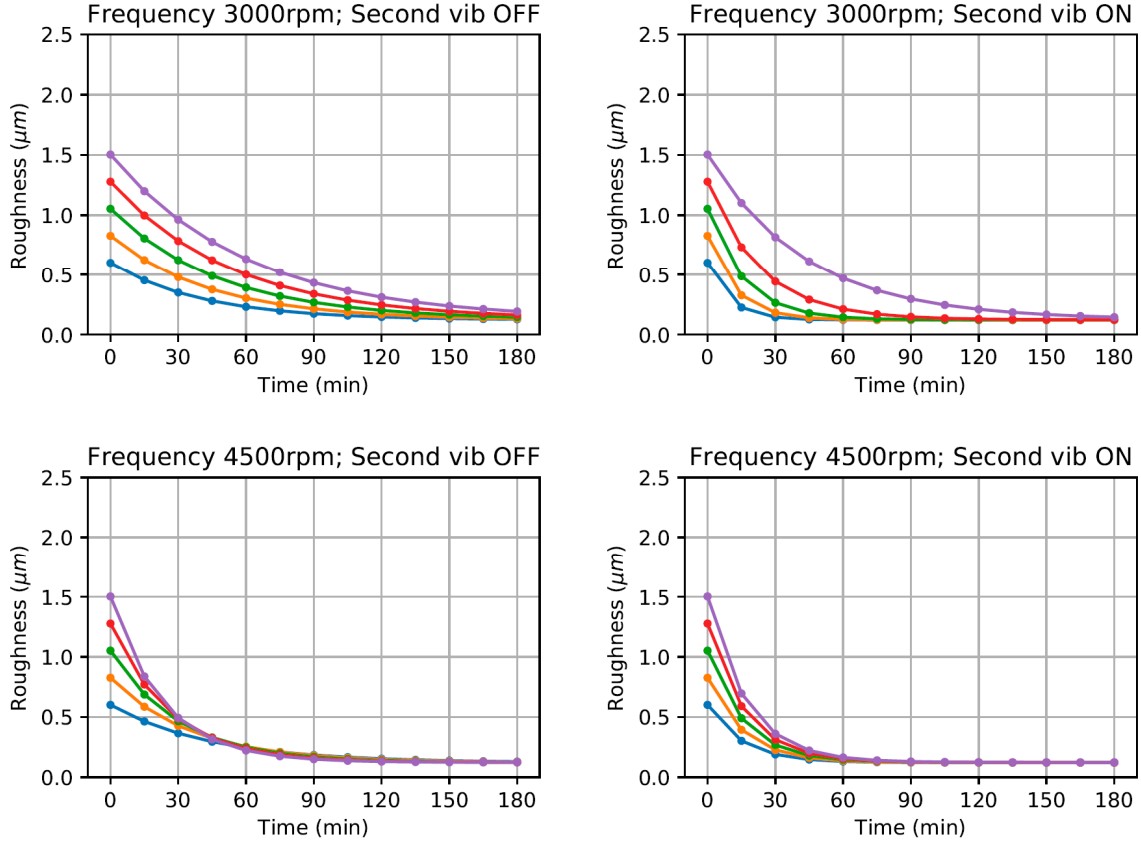

**Figure 9.** Family of curves using exponential regression for bowl system.

## 4. Conclusions

A review of the past and recent modelling methods was done to know the fundamentals of modelling the vibratory finishing process, and how research evolved in trying to solve this complex process more accurately and efficiently. Efforts are applied in either analytical or

mathematical modelling, numerical simulations, computational intelligence, or a combination of these. Though CI models exist for other manufacturing processes such as turning, electroplating, and even in vibropolishing, this study addresses the knowledge gap in the application CI methods in *double-vibropolishing* method, which has great potential in the future of surface finishing technology.

Predictive models were created using three computational intelligence methodologies, namely: linear and non-linear regression, artificial neural networks, and genetic programming. The basic building blocks in constructing the models were discussed, along with preprocessing and preconditioning methodologies.

Results of modelling show that exponential regression was the best CI model for the bowl double-vibropolishing system with a Test MAPE score of 6.1% and a R-squared score of 0.99, while the GP model comes close with Test MAPE and R-squared scores of 5.3% and 0.98 respectively. A family of curves was presented to as a potential tool in predicting surface roughness with time, and hopefully reduce costs in the operation of vibratory finishing machines configured similarly with the bowl system discussed in this work. An example of direct application in the shop floor uses the model to set up the cycle time for each workpiece to be polished depending on initial conditions of the material to be processed, ultimately aimed at eliminating unnecessary processing time.

Further experiment investigations of vibro-polishing process systems can be designed that would include additional variables, different geometry and materials for the workpieces, and different media types. This can bring about robust data analytics in vibratory finishing processes.

**Author Contributions:** J.Y.IIA., K.A. and S.-H.Y.; Methodology, Software, Validation, Formal Analysis, Visualization, and Writing-Original Draft Preparation, J.Y.IIA.; Data curation, K.A and J.Y.IIA.; Supervision, S.-H.Y.; Project Administration, K.A.

**Funding:** This research received no external funding.

**Acknowledgments:** This work was conducted within the Rolls-Royce@NTU Corporate Lab with support from the National Research Foundation (NRF) Singapore under the Corp Lab@University Scheme. The authors would also like to thank Rijul Mediratta and Ketav Majumdar from Rolls-Royce@NTU Corporate Lab; and Thomas Haubold and Kelvin Chan from Rolls-Royce for their contributions.

**Conflicts of Interest:** The authors declare no conflict of interest.

## Nomenclature

| | |
|---|---|
| $R_a$ | Arithmetic average of the roughness profile (μm) |
| $R_i$ | Initial roughness before subjecting to vibropolishing (μm) |
| $R_s$ | Roughness saturation (μm); the minimum roughness that a material achieves when subjected to a vibropolishing process |
| $\hat{R}_a$ | Normalized roughness average (dimensionless) |
| $f_b$ | frequency of vibratory bowl (rpm) |
| $s_v$ | state of vibratory fixture (0:off or 1:on) |
| MAPE | Mean absolute percentage error |

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
