# Peer review of "Predictive Models of Double-Vibropolishing in Bowl System Using Artificial Intelligence Methods"

_jmmp, doi:10.3390/jmmp3010027_

Reviewer 1 Report

Title and thematics of paper is interesting. Article that is in line with the profile of the journal.
The article, however, is illegible in the form presented

Please improve aricle to make it readable

The numbers of the drawings are not quoted in the text,

Incorrect order of content

No description of the experimental conditions found in the final part of the article,

Some results of the research for regression analysis was shown before part of results and discussoion..
After re-writing and organizing the article it is possible to present the advantages of the article.

Author Response

Title and thematics of paper is interesting. Article that is in line with the profile of the journal.
The article, however, is illegible in the form presented

Please improve aricle to make it readable – The authors have ensured due diligence and made sure that the content is easy to understand and is presented in the best way possible
The numbers of the drawings are not quoted in the text, -
This has been done in the revised paper
Incorrect order of content –
The authors have made sure and presented the content in the correct order

No description of the experimental conditions found in the final part of the article, -This has been addressed and can be found in acquisition of data section in the paper

Some results of the research for regression analysis was shown before part of results and discussoion..
After re-writing and organizing the article it is possible to present the advantages of the article.
The article has been presented in a flow easiest to understand. There probably is misinterpretation that the reviewer misinterpreted the exponential extrapolation with the regression model

Reviewer 2 Report

Remarks for the paper:

Predictive models of double-vibropolishing in bowl system using artificial intelligence methods

This paper seems to be well constructed. The state of the art is enough good elaborated. The scientific gap has been identified. This is well planned and done scientific work. The authors applied a methodical apparatus adequate to the assumed goals. Obtained results are interesting from the point of view of the vibropolishing rationalization. But the following comments should be addressed before considering of publication:

1.     Typescript is not prepared enough carefully. For example - what does mean: Error! Reference source not found? besides it, there are some typographical errors.

2.     Table 2 – what does mean “std error”?

3.     I propose to make proper setup (graphical abstract) in methodological part of the paper.

4.     Please emphasize what are the specific conclusions useful for workshop practice.

Author Response

This paper seems to be well constructed. The state of the art is enough good elaborated. The scientific gap has been identified. This is well planned and done scientific work. The authors applied a methodical apparatus adequate to the assumed goals. Obtained results are interesting from the point of view of the vibropolishing rationalization. But the following comments should be addressed before considering of publication:

1.     Typescript is not prepared enough carefully. For example - what does mean: Error! Reference source not found? besides it, there are some typographical errors. – This is due to the possible error with opening the document

2.     Table 2 – what does mean “std error”? – This has been addressed

3.     I propose to make proper setup (graphical abstract) in methodological part of the paper. –The setup is a conventional vibratory bowl that is commonly used in industry and is shown in Figure 1 a). The schematic of double vibropilishing similar to what is used in the current study is shown in Figure 1 b)

4.     Please emphasize what are the specific conclusions useful for workshop practice. – This has been addressed

 Reviewer 3 Report

The reviewer comments of the paper"Predictive models of double-vibropolishing in bowl system using artificial intelligence methods”

- Reviewer

The author presented an article about the prediction model of a double vibrating polishing system using artificial intelligence methods. However, there are several points in the article that require further explanation.

Comment 1:

Abstract as a whole sounds good. However, in the abstract it is necessary to supplement the material of the workpiece for which the experiment was conducted. In addition to qualitative results, it is important to show quantitative results of research. For example, the comparison of errors for the attempted methods of artificial intelligence.

Comment 2:

Introduction should be expanded with a review of articles on vibratory finishing over the past 5 years. At the end of the introduction, you must list the sections of the article, and briefly describe what is done in each section. It is also necessary to clearly state the purpose of the research.

It will also be helpful to include several articles in the introduction:

Robotics and Computer-Integrated Manufacturing 2018, 53, 215-227, doi:10.1016/j.rcim.2018.03.011

Journal of Intelligent Manufacturing 2018, 29(5), 1045-1061, doi:10.1007/s10845-017-1381-8

Comment 3:

Please use the following article structure: Introduction; Materials and methods; Experiment; Results and discussion; Conclusions. Inside these sections use the available subsections.

It is necessary to mention the most significant works in the introduction. It is necessary to clearly show what is the novelty and difference of this work from the existing ones. The number of quotes is enough 25-35 most significant. It is necessary to prescribe a clearer conclusions at the end of the introduction.

Please number the list by reference sequentially. (1, 2, 3, etc.). Using quotes for more than 1-2 quotes in one sentence is not good. Limit yourself to the most important quotes for the topic of the article. Make a more detailed explanation of the quotations in the introduction. For example, 2-4, 5-8, 11-14, 15-16.

Comment 4:

Please discuss the results presented in fig. 3. Sign the names of the horizontal and vertical axes.

Comment 5:

Give a table with the chemical composition of the alloy Ti-6Al-4V. Well in the introduction to show the rationale for consideration in the article is this particular alloy. Show the relevance of this.

Comment 6:

Please discuss the results presented in fig. 3-7. Sign the names of the horizontal and vertical axes.

Comment 7:

Equations 1-4 are original or borrowed from other sources? In the latter case, it is necessary to refer to the source.

Comment 8:

Give a description of the equipment on which the experiment was performed. Indicate the company and country of the manufacturer: machine; tool; measuring et al.

Comment 9:

What is the characteristic used in the article Rs? Give its decoding and description. Give the units, for example, Ra = 0.12.

Comment 10:

In the Discussion of Results section, give a more extensive discussion of the methods discussed in the article. Discuss table 5 and figure 9 in more detail. Now the section looks more like conclusions.

Comment 11:

It will be useful to add a section of Nomenclature in which to sign all the physical quantities encountered in the article. There are many physical quantities in the text and such a section will help to find the description of the necessary element.

For example,

Ra              : Arithmetic average of the roughness profile (µm)

etc.

Comment 12:

The conclusions of the article look vague and inconsistent with the introduction and main work. It is necessary to bring both qualitative and quantitative results of the article. Make a comparison with similar work on this material. Show what is novelty.

Author Response

The author presented an article about the prediction model of a double vibrating polishing system using artificial intelligence methods. However, there are several points in the article that require further explanation.

Comment 1:

Abstract as a whole sounds good. However, in the abstract it is necessary to supplement the material of the workpiece for which the experiment was conducted. In addition to qualitative results, it is important to show quantitative results of research. For example, the comparison of errors for the attempted methods of artificial intelligence. – This has been addressed

Comment 2:

Introduction should be expanded with a review of articles on vibratory finishing over the past 5 years. At the end of the introduction, you must list the sections of the article, and briefly describe what is done in each section. It is also necessary to clearly state the purpose of the research.

It will also be helpful to include several articles in the introduction:

Robotics and Computer-Integrated Manufacturing 2018, 53, 215-227, doi:10.1016/j.rcim.2018.03.011

Journal of Intelligent Manufacturing 2018, 29(5), 1045-1061, doi:10.1007/s10845-017-1381-8

The authors have ensured that all relevant references based on the scope of this study have been included

Comment 3:

Please use the following article structure: Introduction; Materials and methods; Experiment; Results and discussion; Conclusions. Inside these sections use the available subsections. This has been addressed. The experimental section is included as part of the acquisition/pre-processing of data section in the paper

It is necessary to mention the most significant works in the introduction. It is necessary to clearly show what is the novelty and difference of this work from the existing ones. The number of quotes is enough 25-35 most significant. It is necessary to prescribe a clearer conclusions at the end of the introduction. This has been addressed.

Please number the list by reference sequentially. (1, 2, 3, etc.). Using quotes for more than 1-2 quotes in one sentence is not good. Limit yourself to the most important quotes for the topic of the article. Make a more detailed explanation of the quotations in the introduction. For example, 2-4, 5-8, 11-14, 15-16. The authors have ensured that the relevant references have been cited

Comment 4:

Please discuss the results presented in fig. 3. Sign the names of the horizontal and vertical axes. – This has been addressed

Comment 5:

Give a table with the chemical composition of the alloy Ti-6Al-4V. Well in the introduction to show the rationale for consideration in the article is this particular alloy. Show the relevance of this.- This has been addressed

Comment 6:

Please discuss the results presented in fig. 3-7. Sign the names of the horizontal and vertical axes. – The aces for figures 3-7 have been clearly marked and have been discussed in the body of the paper in the relevant section

Comment 7:

Equations 1-4 are original or borrowed from other sources? In the latter case, it is necessary to refer to the source. – The equations that have been borrowed have been duly cited, the ones that are not cited are original and have been developed as part of this study

Comment 8:

Give a description of the equipment on which the experiment was performed. Indicate the company and country of the manufacturer: machine; tool; measuring et al. – Due to the Non Disclosure Agreement that exists between Rolls-Royce and the authors, this information cannot be disclosed.

Comment 9:

What is the characteristic used in the article Rs? Give its decoding and description. Give the units, for example, Ra = 0.12. – This has been addressed

Comment 10:

In the Discussion of Results section, give a more extensive discussion of the methods discussed in the article. Discuss table 5 and figure 9 in more detail. Now the section looks more like conclusions. – This has been addressed and relevant details have been added

Comment 11:

It will be useful to add a section of Nomenclature in which to sign all the physical quantities encountered in the article. There are many physical quantities in the text and such a section will help to find the description of the necessary element. – This has been addressed

For example,

Ra              : Arithmetic average of the roughness profile (µm)

etc.

Comment 12:

The conclusions of the article look vague and inconsistent with the introduction and main work. It is necessary to bring both qualitative and quantitative results of the article. Make a comparison with similar work on this material. Show what is novelty. This has been addressed.